# Nanoscale triboelectrification gated transistor

Tianzhao Bu [1,2,6], Liang Xu [1,2,6], Zhiwei Yang[1,2], Xiang Yang[3], Guoxu Liu[1,2], Yuanzhi Cao[1,2], Chi Zhang [1,2,4✉] & Zhong Lin Wang[1,2,4,5✉]

Tribotronics has attracted great attention owing to the demonstrated triboelectrification-controlled electronics and established direct modulation mechanism by external mechanical stimuli. Here, a nanoscale triboelectrification-gated transistor has been studied with contact-mode atomic force microscopy and scanning Kevin probe microscopy. The detailed working principle was analyzed at first, in which the nanoscale triboelectrification can tune the carrier transport in the transistor. Then with the manipulated nanoscale triboelectrification, the effects of contact force, scan speed, contact cycles, contact region and charge diffusion on the transistor were investigated, respectively. Moreover, the manipulated nanoscale triboelectrification serving as a rewritable floating gate has demonstrated different modulation effects by an applied tip voltage. This work has realized the nanoscale triboelectric modulation on electronics, which could provide a deep understanding for the theoretical mechanism of tribotronics and may have great applications in nanoscale transistor, micro/nano-electronic circuit and nano-electromechanical system.

[1] CAS Center for Excellence in Nanoscience, Beijing Key Laboratory of Micro-nano Energy and Sensor, Beijing Institute of Nanoenergy and Nanosystems, Chinese Academy of Sciences, 100083 Beijing, China. [2] School of Nanoscience and Technology, University of Chinese Academy of Sciences, 100049 Beijing, China. [3] Institute of Semiconductors, Chinese Academy of Sciences, 100083 Beijing, China. [4] Center on Nanoenergy Research, School of Physical Science and Technology, Guangxi University, 530004 Nanning, China. [5] School of Material Science and Engineering, Georgia Institute of Technology, Atlanta, GA 30332-0245, USA. [6] These authors contributed equally: Tianzhao Bu, Liang Xu. ✉email: czhang@binn.cas.cn; zlwang@gatech.edu

With the development of micro/nano-technology, electronics are expected to be more miniaturized[1–3], multifunctional[4–6], and intelligent[7–9] to promote the information technology advancement. By integrating micro/nano-electronics into multifunctional micro/nano-systems and large-scale networks for environmental monitoring[10–12], human-machine interfacing[13–15], biomedical diagnosis/therapy[16–19], and so on, everything could be connected to the Internet for information interaction and intelligent identification in the future[20]. However, the interaction mechanisms between most of the micro/nano-electronics and human/environment are passive so far, which greatly limits the information acquisition[21]. Therefore, it is highly desired for micro/nano-electronics to establish active interactions with the external stimuli to achieve direct information acquisition.

Triboelectric nanogenerator (TENG) as an emerging energy technology has been invented by Wang in 2012[22,23], which could convert a variety of mechanical motions into electrical outputs based on the conjugation effects of triboelectrification and electrostatic induction[24–28]. In recent years, by coupling the triboelectricity and semiconductor properties, tribotronics, as an emerging research field has been proposed, which has demonstrated various triboelectrification-controlled electronics and established direct modulation mechanism by external mechanical stimuli[29–32]. However, the interactive interfaces between external environment and electronics in current tribotronic devices are all in the macro scale, which has limited the integration and modularization of tribotronics. When the size scales down to the micro or nano range, whether the modulation effect still exists is a critical question for tribotronics.

In this work, a nanoscale triboelectrification-gated transistor (NTT) has been studied with contact-mode atomic force microscopy (AFM) and scanning Kevin probe microscopy (SKPM). The detailed working principle was analyzed at first, in which the nanoscale triboelectrification generated on the top dielectric layer by AFM tip can tune the carrier transport in the NTT. Then with the manipulated nanoscale triboelectrification by AFM, the effects of contact force, scan speed, contact cycles, contact region and charge diffusion on the characteristics of the NTT were investigated, respectively. Moreover, the manipulated nanoscale triboelectrification serving as a rewritable floating gate has demonstrated different modulation effects on the NTT by an applied tip voltage. This work has experimentally realized the nanoscale triboelectric modulation on transistor by using AFM and demonstrated micro/nano-scale tribotronics, which could provide a deep understanding for the theoretical mechanism of tribotronics. The implementation of the NTT can provide direct interactions of electronics with external stimuli, which is highly desired for the development of micro/nano-electronics in diversity and functionality. This may have great prospects in nanoscale transistor, micro/nano-electronic circuit and nano-electromechanical system (NEMS) for human-machine interfacing, flexible electronics, biomedical diagnosis/therapy and so on.

## Results

**Mechanism of the NTT.** As schematically illustrated in Fig. 1a, the basic structure of NTT is similar to the traditional metal oxide semiconductor field-effect transistor (MOSFET) without the top gate electrode. And, the top silicon dioxide ($SiO_2$) layer is rubbed by an AFM tip which has a radius of 20–40 nm (Fig. 1b) to ensure the contact area is in nanoscale during the process of triboelectrification. The scanning electron microscope (SEM) images of the NTT in cross-sectional view are shown in Fig. 1c. The test method for NTT is stated as follows. First, the channel region ($5 \times 5$ μm) of the NTT is located with tapping-mode AFM

(Fig. 1d). Then, the AFM is switched into the contact mode to scan and rub the top $SiO_2$ surface above the channel region for nanoscale triboelectrification (Fig. 1e). Subsequently, the nanoscale triboelectrification could be quantitatively characterized by measuring the potential distribution of the top $SiO_2$ surface using SKPM (Fig. 1f, g). The electrical characteristics of the NTT are constantly monitored during the whole process to obtain the nanoscale triboelectrification tuned properties.

The working principle of the NTT is illustrated with cross-sectional schematics in Fig. 1h, which could be mainly divided into two stages. During the process of nanoscale triboelectrification, the top $SiO_2$ surface above the channel region is scanned and rubbed by the AFM tip in contact mode. Based on the electron transfer mechanism, electrons from the AFM tip could transfer onto the top $SiO_2$ surface, because the AFM tip used in this work is silicon (Si) probe, which has a smaller effective work function than $SiO_2$. After contact procedure, considerable transferred charges are bound on the top $SiO_2$ surface, resulting in an inner electric field at the channel region, which can attract holes and repel electrons in the P-type silicon (P-Si) layer. Therefore, the concentration of holes as majority carriers at the channel region is enhanced, leading to the formation of enhancement zone, which could increase the drain current $I_d$ of NTT.

The energy band diagrams in the vertical direction of the channel region are also utilized to analyze the working mechanism of NTT. As shown in Supplementary Fig. 1, W is the effective work function of the Si probe, $E_0$ is the difference between vacuum level $E_{vcc}$ and the highest filled surface energy states of $SiO_2$. Thus, the Fermi level of the Si probe is above the highest filled surface energy states of $SiO_2$ in the initial state (Supplementary Fig. 1a). Meanwhile, we define a threshold distance $z$ below which electrons could transfer between two surfaces, and above $z$ the barrier is large enough to prevent any tunneling. After contact (Supplementary Fig. 1b), electrons from the Si probe have flowed to the top $SiO_2$ to fill up the surface states as high as the Fermi level of Si. At this time, a built-in electrical field applied on the $SiO_2$ and P-Si layers is formed due to the transferred electrons on the top $SiO_2$ surface, which can upward bend the energy band of P-Si at the interface. Therefore, the concentration of holes in the valence band of P-Si is enhanced, which could induce the increase of drain current.

**Characteristics of the NTT.** The electrical properties of the transistor are first measured by applying an external bottom gate voltage $V_{bg}$ (Supplementary Fig. 2a). The $I_d$–$V_{bg}$ transfer characteristics of the transistor with different $V_{bg}$ from −30 V to 0 are shown in Supplementary Fig. 2b, which indicate that the drain current monotonously increases with increasing the negative bottom gate voltage, when the negative bottom gate voltage is more than around −17 V. And, drain current has the maximum enhancing speed at a bottom gate voltage of −22 V. Therefore, in order to make the drain current of NTT more sensitive to the nanoscale triboelectrification, a $V_{bg}$ of −22 V is applied in the following experiments for optimizing test method. The equivalent test circuit diagram is shown in Supplementary Fig. 3.

The effect of contact force $F$ on the characteristics of the NTT is first investigated. In this experiment, the top $SiO_2$ surface above the channel region was scanned by the AFM tip once at a scan speed of 10 μm/s. As shown in Supplementary Fig. 4a-c, the changed top $SiO_2$ surface potential above the channel region $\Delta V$ is enhanced with the contact force increasing from 0 to 2.5 nN, and approximately reaches a saturated value of −1.48 V when the contact force is larger than 2 nN. The corresponding drain current $I_d$ of the NTT shows a similar trend following the increase of contact force, which is enhanced from 139 μA to 272 μA and

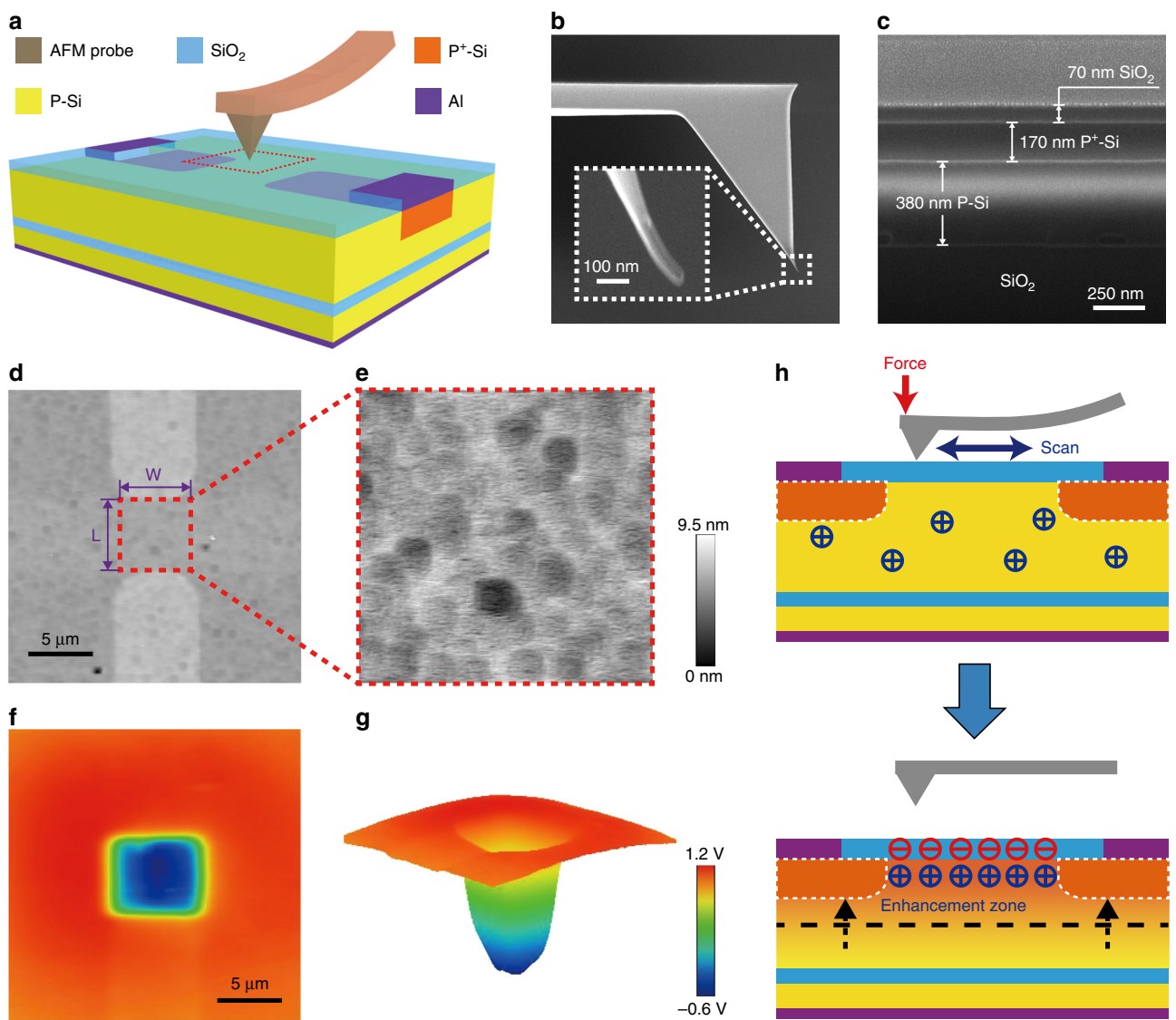

**Fig. 1 Overview of the nanoscale triboelectrification-gated transistor (NTT). a** Schematic illustration of the NTT gated by nanoscale triboelectrification. Scanning electron microscope (SEM) images of the (**b**) atomic force microscopy (AFM) tip and (**c**) NTT in cross-sectional view. AFM images of (**d**) the top SiO₂ surface in tapping mode and (**e**) the rubbed area above the channel region in contact mode. **f** Surface potential distribution of the top SiO₂ layer after regionally rubbed by the AFM tip measured in scanning Kevin probe microscopy (SKPM) mode. **g** 3D image of the measured surface potential distribution. **h** Schematic working principle of the NTT gated by nanoscale triboelectrification.

reaches saturation when contact force is larger than 2 nN (Supplementary Fig. 4d–e). During the process of nanoscale triboelectrification, the increasing contact force can decrease the potential barrier and induce more electrons to transfer onto the SiO₂ surface, until reaching the saturation state when the highest filled surface energy state of SiO₂ is as high as the Fermi level of Si. In Supplementary Fig. 4, as the contact force increases in the low range, more charges are transferred that leads to the rise of the highest filled surface energy state of SiO₂. When the contact force increases to 2 nN, the amount of transferred charges is large enough to make the highest filled surface energy state of SiO₂ almost as high as the Fermi level of Si. So, the contact force larger than 2 nN will hardly lead to more transferred charges, and the KPFM contrast shows the fast saturation.

Then we have investigated the effect of scan speed $v$ on the characteristics of the NTT. In this experiment, the top SiO₂ surface above the channel region was scanned by the AFM tip once at a contact force of 1 nN. The changed top SiO₂ surface potential above the channel region $\Delta V$ is enhanced as the scan

speed decreases from 16 μm s⁻¹ to 2 μm s⁻¹, and reaches saturation when scan speed decreases to 4 μm s⁻¹ (Supplementary Fig. 5a–c). The corresponding $I_d$ output characteristics also shows an increasing trend with the decreasing of scan speed, and reaches a saturated value of 272 μA when the scan speed is less than 4 μm s⁻¹ (Supplementary Fig. 5d–e). A possible reason is that the decreasing scan speed could allow better contact between the AFM tip and the SiO₂ surface until a very low speed, enabling more charges to be transferred onto the SiO₂ surface. In addition, the contact force and scan speed are fixed to 1 nN and 10 μm s⁻¹, respectively, in all the next experiments.

According to the previous work[33], the transferred charge density $\sigma$ on the dielectric layer surface is related to contact cycles $n$, which can be described by Eq. 1:

$$\sigma = \sigma_0 \exp\left(-\frac{n}{n_0}\right) + \sigma_\infty \left[1 - \exp\left(-\frac{n}{n_0}\right)\right] \quad (1)$$

where $\sigma_0$ is the surface charge density before contact; $\sigma_\infty$ is the saturated surface charge density after multiple contacts; $n_0$ is the

saturation constant that controls the saturation rate. Moreover, based on the parallel plate capacitance model, the changed top $SiO_2$ surface potential above the channel region $\Delta V$ could be determined by Eq. 2:

$$\Delta V = \sigma S \times \frac{d_{SiO_2}}{\varepsilon_0 \varepsilon_{SiO_2} S} = \frac{\sigma d_{SiO_2}}{\varepsilon_0 \varepsilon_{SiO_2}} \quad (2)$$

where $d_{SiO2}$ is the thickness of $SiO_2$, $\varepsilon_0$ and $\varepsilon_{SiO2}$ are the vacuum dielectric constant and the relative dielectric constant of $SiO_2$, $S$ is the charged area. In Eq. 2, $\Delta V$ is not related to $S$, which means that for the capacitive devices, the voltage generated by triboelectric charges is independent of area. This is meaningful to the miniaturization of tribotronics. Combining Eqs. 1 and 2, the relation between $\Delta V$ and $n$ is shown in Eq. 3:

$$\Delta V = V_0 \exp\left(-\frac{n}{n_0}\right) + V_\infty\left[1 - \exp\left(-\frac{n}{n_0}\right)\right] \quad (3)$$

where $V_0$ is the top $SiO_2$ surface potential above the channel region before contact, $V_\infty$ is the saturated surface potential after multiple contacts. As for the NTT, the top gate voltage is equivalent to the top $SiO_2$ surface potential above the channel region:

$$V_{tg} = \Delta V + V_0 \quad (4)$$

Therefore, in the ideal state the relation between $I_d$ and $\Delta V$ could be described by Eq. 5:

$$I_d = \frac{W}{L}\frac{\varepsilon_0 \varepsilon_{SiO_2}}{d_{SiO_2}}\mu_p\left(-\Delta V - V_0 - V_t - \frac{V_d}{2}\right)V_d \quad (5)$$

where $W$ and $L$ are the channel width and length of the NTT, $\mu_P$ is the hole mobility, $V_t$ is the threshold voltage of the NTT. Combining Eqs. 3 and 4, the drain current of the NTT can be determined by Eq. 6:

$$I_d = \frac{W}{L}\frac{\varepsilon_0 \varepsilon_{SiO_2}}{d_{SiO_2}}\mu_p\left\{V_0 \exp\left(\frac{n}{n_0}\right) - V_\infty\left[1 + \exp\left(\frac{n}{n_0}\right)\right] - V_0 - V_t - \frac{V_d}{2}\right\}V_d \quad (6)$$

The corresponding experimental results are shown in Fig. 2. First, the potential distributions of the top $SiO_2$ surface after regionally rubbed by the AFM tip with increasing contact cycles are measured (Fig. 2a). Fig. 2b shows the measured potential distributions in cross-sectional view. And, Fig. 2c plots potential differences between the rubbed and surrounding area with different contact cycles, which fit Eq. 3 very well. By the synchronous electrical measurement of the NTT, drain current continuously increases from 139 μA to 257 μA during the process of nanoscale triboelectrification, as shown in Supplementary Fig. 6. Fig. 2d shows the $I_d$ output characteristics at a drain voltage of 5 V with increasing contact cycles from 0 to 3. And, Fig. 2e plots $I_d$–$n$ transfer characteristics of the NTT, which has a good aggrement with Eq. 6. The $I_d$–$V_d$ output characteristics with different contact cycles are shown in Fig. 2f. The experimental results indicate that the drain current is enhanced with increasing contact cycles and reaches to a saturation value of 280 μA, when the contact cycles is more than 2. The increasing contact cycles can induce more transferred charges and leads to the rise of the highest filled surface energy state of $SiO_2$. According to Fig. 2, when the $SiO_2$ surface is scanned twice by the AFM Si tip, the amount of transferred charges is large enough to make the highest filled surface energy state of $SiO_2$ almost as high as the Fermi level of Si. So, the increasing contact cycles (more than 2) will hardly lead to more transferred charges, and the KPFM contrast shows the fast saturation.

Based on the test method for NTT, the contact region on the top $SiO_2$ surface can be accurately controlled by AFM. This

provides a direct method to study the influence of charged area of the top $SiO_2$ surface on the electrical properties of NTT. As shown in Fig. 3a and Fig. 3b, the charged area can be effectively manipulated by changing the length $l$ and width $w$ of the contact region, respectively. The $I_d$ output characteristics at a drain voltage of 5 V with different contact lengths from 0 to 5 μm are shown in Fig. 3c. And, Fig. 3d depicts $I_d$–$l$ transfer curve of the NTT. Fig. 3e shows the $I_d$–$V_d$ output characteristics with different contact lengths. The experimental results indicate that as the contact length increases the drain current of the NTT is enhanced. Similar variation trend of drain current with increasing contact widths is shown in Fig. 3f–h. The difference is that the increasing rate of drain current with contact length is less than that with contact width in small region. This is because when the contact width is changed, the source and drain electrodes of the NTT remain connected by the charged area, thus the carrier transport channel always exists. By contrast, when the contact length is less than the channel length, the source and drain electrodes of the NTT are not connected by the charged area anymore, so the complete enhanced carrier transport channel no longer exists. Therefore, compared with the contact length, the contact width is more efficient in tuning the carrier transport of the NTT.

The transferred charges from the AFM tip cannot be bounded on the top $SiO_2$ surface forever, and the surface diffusion of electrons is considered to exist. So, we have studied the effect of charge diffusion on the characteristics of the NTT in this work. The top $SiO_2$ surface above the channel region is first scanned twice by the AFM tip in contact mode to ensure that the amount of transferred charges reaches saturation. Then the potential distribution of the top $SiO_2$ surface and the electrical properties of the NTT are measured every 2 h. The potential distributions of the top $SiO_2$ surface with different dissipation times $t$ are shown in Fig. 4a. Fig. 4b depicts the corresponding potential distributions in cross-sectional view, which indicate that, as the dissipation time increases, the absolute value of potential difference between the rubbed and surrounding area is decreased and the area of charged region becomes larger at the same time. The synchronously measured $I_d$ output characteristics of the NTT at a drain voltage of 5 V with different dissipation times from 0 to 12 h are shown in Fig. 4c. And, Fig. 4d plots $I_d$–$t$ transfer curve. Fig. 4e shows the $I_d$–$V_d$ output characteristics of the NTT with different dissipation times. The experimental results indicate that with increasing dissipation time, the drain current of the NTT is reduced, and the reduction rate of the drain current is decreased. This is because as the dissipation time increases, the absolute value of potential difference between the rubbed and surrounding area is decreased, which is equivalent to the decreasing of negative top gate voltage, resulting in the reduction of drain current.

**The rewritable floating gate for the NTT.** The nanoscale triboelectrification has been demonstrated to be effectively manipulated by an applied electric field. In order to realize a variety of modulation effects on the electrical properties of the NTT, a tip voltage $V_T$ is applied during the process of nanoscale triboelectrification (Fig. 5a).

The energy band diagrams in the vertical direction of the channel region are utilized to illustrate the working mechanism of NTT with an applied tip voltage (Supplementary Fig. 7). When the tip voltage is negative ($V_T < 0$), which can raise the Fermi level of Si probe, more electrons are transferred from the AFM tip to the top $SiO_2$ layer to fill up the higher surface energy states. Thus, the built-in electric field applied on the top $SiO_2$ and P-Si layers is enhanced, and the energy band of P-Si bends further upwards in the interface, leading to further enhancement of hole

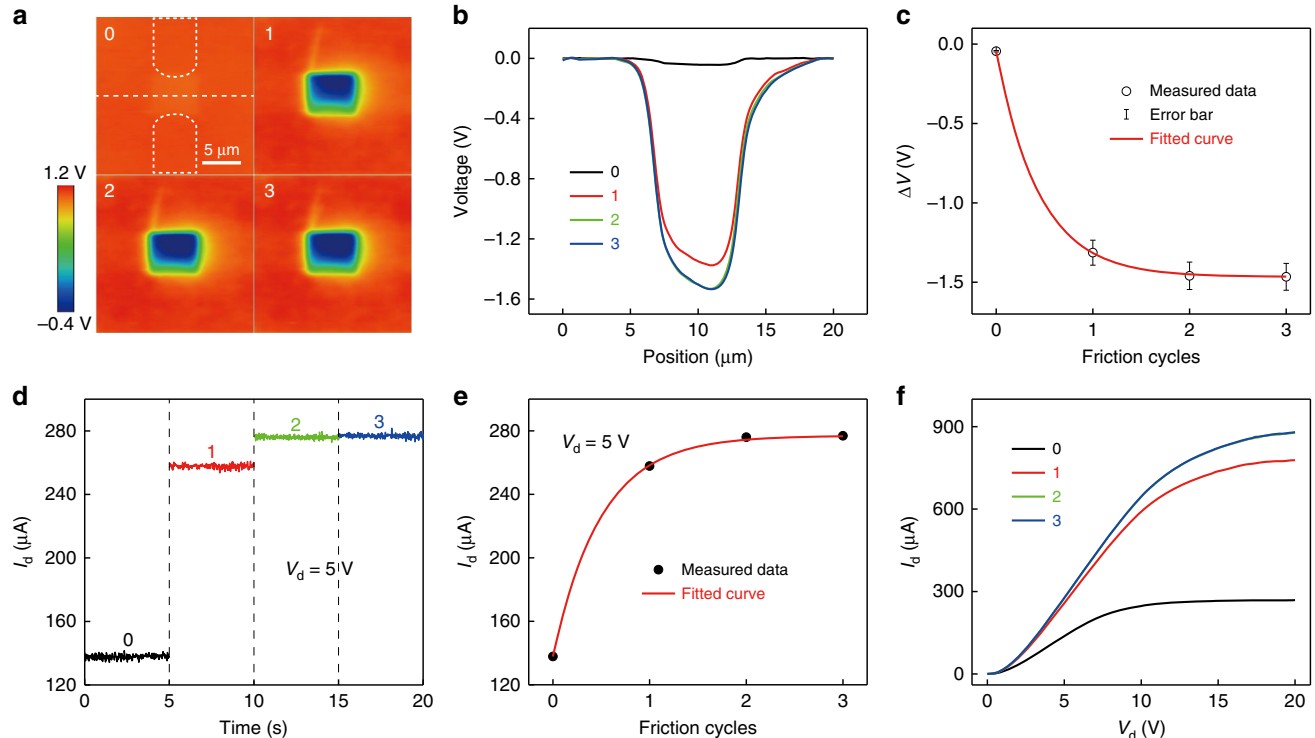

**Fig. 2 The effect of contact cycles on the characteristics of the NTT. a** Surface potential distribution of the NTT after regionally rubbed by the AFM tip with increasing contact cycles. **b** The corresponding potential distribution in cross-sectional view. **c** The potential difference between the rubbed and surrounding area with increasing contact cycles. **d** $I_d$ output characterisics at a drain voltage of 5 V with different contact cycles from 0 to 3. **e** The $I_d$–$n$ transfer characteristics. **f** $I_d$–$V_d$ output characteristics with different contact cycles. All error bars in the figure represent s.d. of the data.

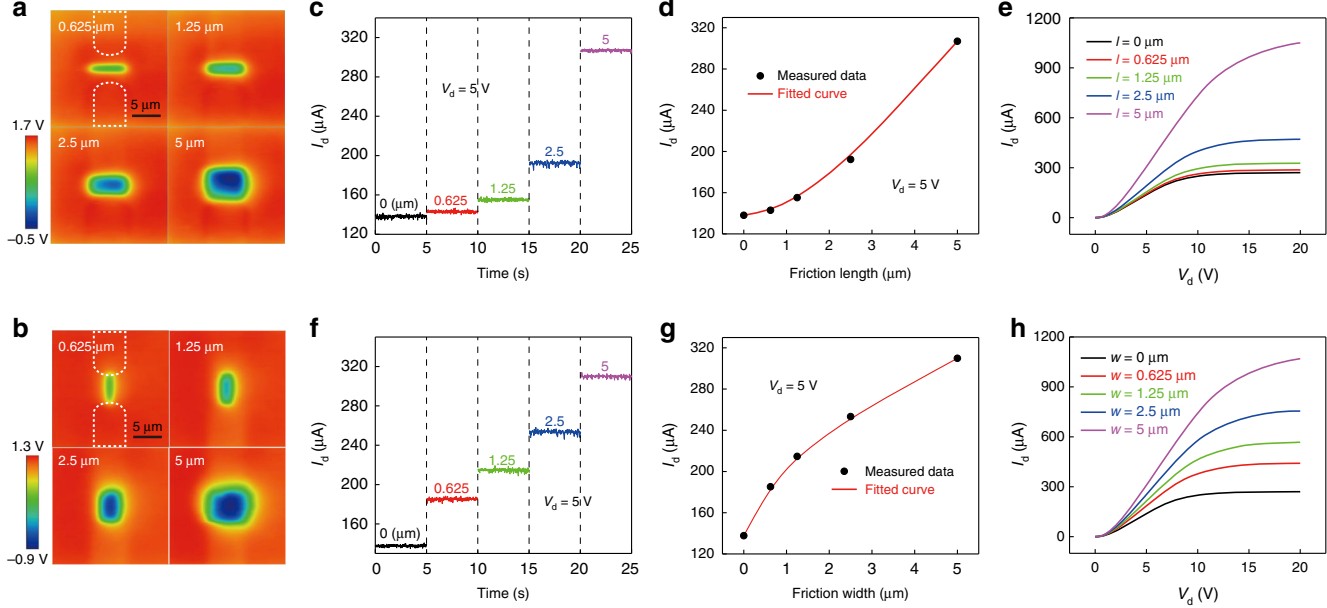

**Fig. 3 The effect of the contact region on the characteristics of the NTT.** Surface potential distributions of the NTT after regionally rubbed by the AFM tip with the increasing (**a**) length and (**b**) width. **c** $I_d$ output characterisics at a drain voltage of 5 V with different contact lengths from 0.625 μm to 5 μm. **d** The $I_d$–$l$ transfer characteristics. **e** $I_d$–$V_d$ output characteristics with the different contact lengths. **f** $I_d$ output characterisics at a drain voltage of 5 V with different contact widths from 0.625 μm to 5 μm. **g** The $I_d$–$w$ transfer characteristics. **h** $I_d$–$V_d$ output characteristics with different contact widths.

concentration in valence band and drain current of the NTT. On the contrary, a positive tip voltage can lower the Fermi level of the Si probe, reducing the number of electrons flowed from the AFM tip to the top $SiO_2$ surface. As the positive tip voltage increases to a certain value ($V_T = V_N$), the Fermi level of Si probe can be as low as the highest filled surface states of $SiO_2$. At this condition, no charge can flow between the AFM tip and the top $SiO_2$ surface. When the tip voltage is more positive than the nullified voltage ($V_T > V_N$), electrons can flow reversely from the top $SiO_2$ surface to the AFM tip. Therefore, a positive built-in electric field is

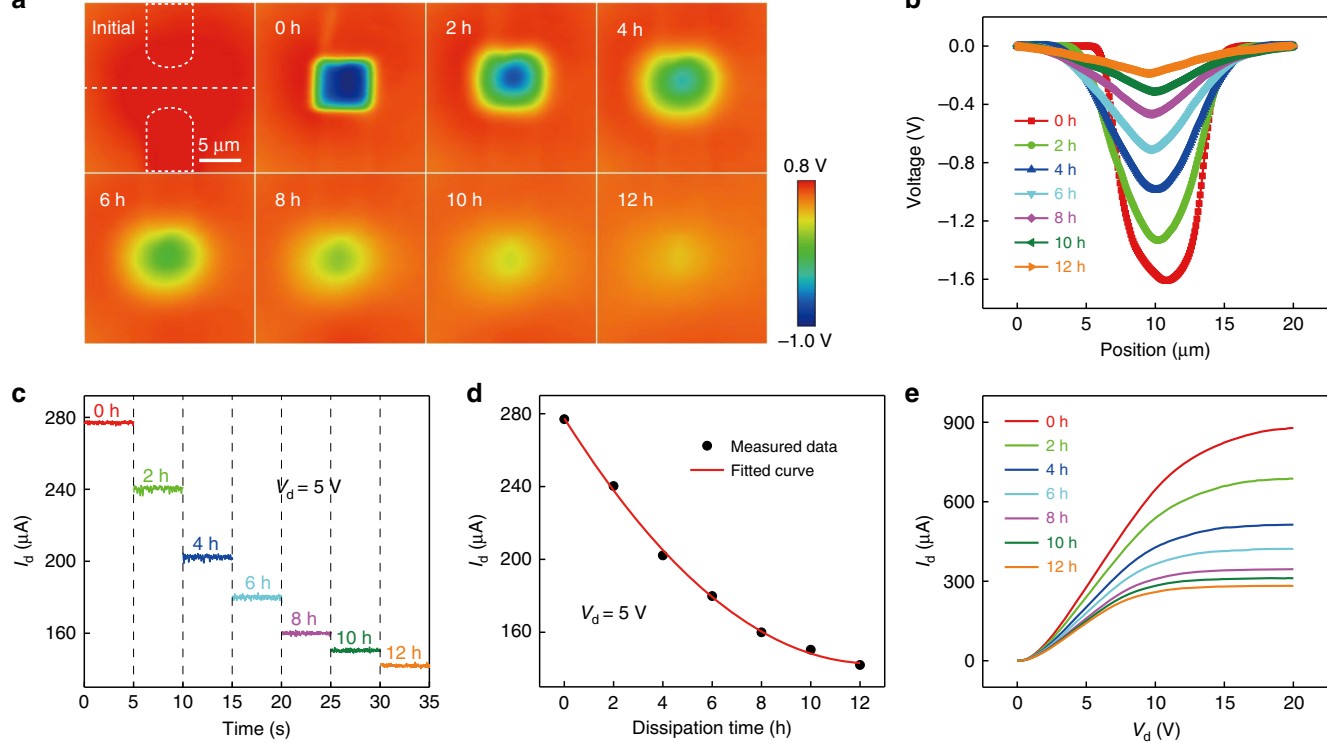

**Fig. 4 The effect of charge diffusion on the characteristics of the NTT. a** Surface potential distributions of the NTT after regionally rubbed by the AFM tip with increasing dissipation time. **b** Corresponding potential distributions in cross-sectional view with increasing dissipation time. **c** $I_d$ output characterisics at a drain voltage of 5 V with different dissipation times from 0 to 12 h. **d** The $I_d$–$t$ transfer characteristics. **e** $I_d$–$V_d$ output characteristics with different dissipation times.

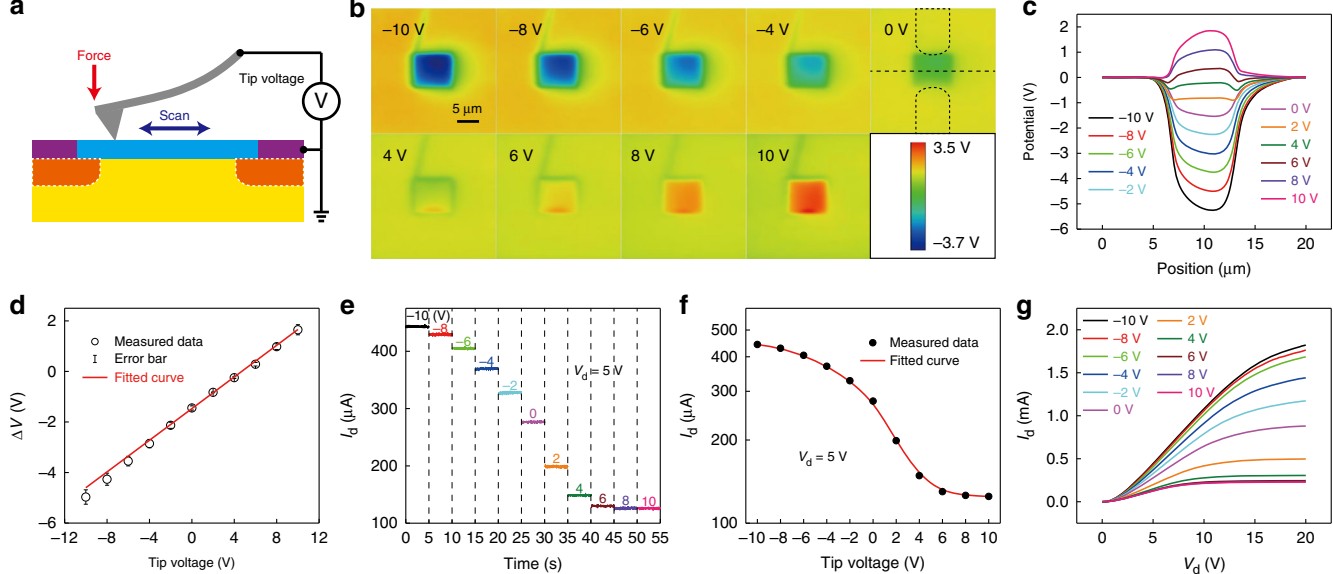

**Fig. 5 The rewritable floating gate for the NTT by applying a tip voltage. a** Schematic illustration of applying a tip voltage during the process of nanoscale triboelectrification. **b** Surface potential distributions of the NTT after regionally rubbed by the AFM tip with different tip voltages. **c** Corresponding potential distributions in cross-sectional view with different tip voltages. **d** The potential difference between the rubbed and surrounding area with different tip volltages. **e** $I_d$ output characterisics at a drain voltage of 5 V with different tip voltages from −10 V to 10 V. **f** The $I_d$–$V_T$ transfer characteristics. **g** $I_d$–$V_d$ output characteristics with different tip voltages.

applied on the top $SiO_2$ and P-Si layers, resulting in downwards bend of the energy band of P-Si in the interface, so the hole concentration in valence band and drain current of the NTT are decreased.

According to the previous work[34], the transferred charge density $\sigma$ can be modulated by the tip voltage $V_T$, which can be described by Eq. 7:

$$\sigma = \frac{V_T + [(W - E_0)/e]\left(1 + d_{SiO_2}/\varepsilon_{SiO_2}z\right)}{d_{SiO_2}/\varepsilon_0\varepsilon_{SiO_2} + (1/\bar{N}_s(E)e^2)\left(1 + d_{SiO_2}/\varepsilon_{SiO_2}z\right)} \quad (7)$$

where $e$ is the elementary charge, and $\overline{N}_s(E)$ is defined as the averaged surface density of states. Combining Eqs. 2 and 6, we have

$$\Delta V = \frac{V_T + [(W - E_0)/e]\left(1 + d_{SiO_2}/\varepsilon_{SiO_2}z\right)}{1 + [1/\bar{N}_s(E)e^2]\left(1 + d_{SiO_2}/\varepsilon_{SiO_2}z\right)\left(\varepsilon_0\varepsilon_{SiO_2}/d_{SiO_2}\right)} \quad (8)$$

Further combining Eqs. 5 and 8, the relation between the drain current of the NTT and the tip voltage can be determined by Eq. 8:

$$I_d = \frac{W}{L}\frac{\varepsilon_0\varepsilon_{SiO_2}}{d_{SiO_2}}\mu_p\left\{\frac{-V_T - [(W - E_0)/e]\left(1 + d_{SiO_2}/\varepsilon_{SiO_2}z\right)}{1 + [1/\bar{N}_s(E)e^2]\left(1 + d_{SiO_2}/\varepsilon_{SiO_2}z\right)\left(\varepsilon_0\varepsilon_{SiO_2}/d_{SiO_2}\right)} - V_0 - V_t - \frac{V_d}{2}\right\}V_d \quad (9)$$

This equation means that the drain current of the NTT can be effectively tuned in different ways by the tip voltage.

In the experiments, the effect of contact cycles on the characteristics of the NTT with an applied tip voltage is first measured. As shown in Supplementary Fig. 8, the amount of transferred charges on the top $SiO_2$ surface and the drain current of the NTT saturate after rubbed by the AFM tip once at a tip voltage of −2 V. The potential distributions of the top $SiO_2$ surface with increasing tip voltage are shown in Fig. 5b. Fig. 5c depicts the corresponding potential distributions in cross-sectional view. The potential difference between the rubbed and surrounding area almost linearly increases with the tip voltage, as shown in Fig. 5d. The $I_d$ output characteristics with different tip voltages from −10 V to 10 V are shown in Fig. 5e. And, Fig. 5f plots the $I_d$–$V_T$ transfer characteristics of the NTT. The transfer curve of the NTT is saturated with the $I_d$ of 125 μA in the positive direction, and in the negative direction, the drain current shows a saturation trend and is almost saturated with a value of 443 μA. Fig. 5g shows the $I_d$–$V_d$ output characteristics with different tip voltages. Meanwhile, the modulation of $\Delta V$ and $I_d$ by the tip voltage are repeatable, which demonstrates a rewritable floating gate for the NTT. In addition, we have fabricated a top metal gate electrode on the transistor (Supplementary Fig. 9a-b). The corresponding $I_d$–$V_{tg}$ transfer curve is shown in Supplementary Fig. 9c, from which we can calculate the mobility ($\mu = \frac{Lg_m}{WC_{ox}V_d}$, where $\mu$, $g_m$, $L$, $W$ and $C_{ox}$ are the mobility, transconductance, channel length, channel width and gate oxide capacitance), $I_{on/off}$ ratio, and $V_{th}$ as 193 cm$^2$ V$^{-1}$ s$^{-1}$, 23 and 2.8 V, respectively. As for the NTT, the changed top $SiO_2$ surface potential above channel region $\Delta V$ is considered as the actual voltage that applied on the top gate of the NTT by the AFM tip. Thus, we get the $I_d$–$\Delta V$ transfer curve of the NTT to compare with the top metal gate transistor. According to the $I_d$–$V_T$ curve shown in Fig. 5f, the $I_d$–$\Delta V$ transfer curve of the NTT can be plotted as shown in Supplementary Fig. 9d, in which each $\Delta V$ corresponds to a tip voltage $V_T$ (Fig. 5d). The mobility, $I_{on/off}$ ratio, and $V_{th}$ of the NTT can be further calculated as 431 cm$^2$ V$^{-1}$ s$^{-1}$, 4 and 1.1 V, respectively. As a contrast, there are a few differences on the mobility, $I_{on/off}$ ratio, and $V_{th}$ between top metal gate transistor

and the NTT due to the stray capacitance of top metal gate, the residual charge after nanoscale triboelectrification and so on. However, both sets of values are comparable, which indicate that the top metal gate electrode can be replaced by nanoscale triboelectrification-gating. The effect of charge diffusion on the characteristics of the NTT with an applied tip voltage is also studied. The top $SiO_2$ surface above the channel region is first scanned once by the AFM tip in contact mode at a tip voltage of −2 V. Then the potential distribution of the top $SiO_2$ surface and the electrical properties of the NTT are measured every hour. As shown in Supplementary Fig. 10, the absolute value of potential difference between the rubbed and surrounding area is decreased and the area of charged region becomes larger with rising dissipation time. At the same time, the drain current of the NTT is reduced from 312 μA to 265 μA within 8 h.

## Discussion

In summary, we have studied the tribotronic transistor gated by nanoscale triboelectrification with contact-mode AFM and SKPM. The detailed working principle was analyzed at first, in which the nanoscale triboelectrification generated on the top dielectric layer by AFM tip can tune the carrier transport in the NTT in a space of less than a few micrometers. Then, the effects of contact force, scan speed, contact cycles, contact region and charge diffusion on the characteristics of the NTT were investigated, and the results have shown a good agreement with the theoretical analysis. Moreover, the controllable nanoscale triboelectrification by an applied tip voltage can serve as a rewritable floating gate, which has demonstrated different modulation effects on the NTT. This work has realized the nanoscale triboelectric modulation on electronics, which could provide a deep understanding for the theoretical mechanism of tribotronics and may have great applications in nanoscale transistor, micro/nano-electronic circuit and NEMS.

## Methods

**Fabrication of the NTT.** The detailed fabrication process of the NTT is shown in Supplementary Fig. 11. First, an SOI wafer with the sandwich structure was prepared, which has a top 380 nm thick P-Si layer, a buried 500 nm thick $SiO_2$ layer and a bottom 500 μm thick P-Si. Then, part of the P-type channel layer was masked by lithography (M1) (SUSS MA/BA 6), and two heavily doped P$^+$-Si areas were formed by Boron ion implantation (Varian 350). A layer of $SiO_2$ with a thickness of 70 nm was formed by thermal oxidization (L4514II2F) on the top and bottom surfaces. The top $SiO_2$ surface was next masked by lithography (M2), and the electrode areas were opened by inductively coupled plasma (ICP) etching the top $SiO_2$ layer (STS Multiplex AOE). The top surface was masked again by lithography (M2), and two Aluminum (Al) pads with a thickness of 1 μm were deposited on the surface of P$^+$-Si areas with Ohmic contacts by RF sputtering (Denton Discovery635) as the drain and source electrodes. At last, the bottom $SiO_2$ layer was etched and a 1 μm thick Al layer was deposited on the surface of bottom P-Si layer serving as the bottom gate electrode.

**Characterization of the NTT.** First, Si probe was prepared by mechanically removing the Platinum (Pt) layer of the conductive probe (AC240TM from Olympus). More specifically, the probe was scanned on a sapphire film with a contact force of 600 nN for three cycles. The morphology of Si probe was characterized by Hitachi SU8020 with an acceleration voltage of 7 kV. Then a standardized focused ion beam (FIB) method was used to etch the channel region of the NTT (FEI Helios 600I), so that the cross-sectional view of the NTT can be characterized as shown in Supplementary Fig. 12. The measurement platform for the NTT was conducted in an MFP-3D AFM from Asylum Research at ambient environment with the relative humidity of 20–30%. In contact mode, the sensitivity of optical lever and spring constant of the cantilever are 197 nm V$^{-1}$ and 0.9 N m$^{-1}$, respectively. And the deflection of cantilever was set to be 0.005 V. Thus, the contact force was controlled as 1 nN during the process of nanoscale triboelectrification. The synchronous $I_d$–$V_d$ output characteristics of the NTT were performed by using a Keithley 4200 A semiconductor characterization system. And the transfer characteristics were measured by using an SR570 low-noise current amplifier (Stanford Research System) and a programmable DC power supply (RIGOL DP832).

## Data availability
All data needed to evaluate the conclusions in the paper are present in the paper and/or the Supplementary Information. Additional data related to this paper may be requested from the authors. The source data underlying all figures can be found in the Source Data file.

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

## Acknowledgements
The authors thank the support of the National Natural Science Foundation of China (Nos. 51922023, 51605033, 61874011), Beijing Natural Science Foundation (No. 4192070).

## Author contributions
T.B., L.X., C.Z., and Z.L.W. conceived the idea and designed the experiment. T.B. and L.X. carried out the nanoscale triboelectrification-gated transistor experiments. Z.Y. and X.Y. contributed to the device fabrication. G.L. and Y.C. helped with the data analysis. T.B., L.X., and C.Z. wrote the paper. All the authors discussed the results and commented on the paper.

## Competing interests
The authors declare no competing interests.
