## [Peer Review File · Nature Communications]

Reviewers' comments:

Reviewer #1 (Remarks to the Author):

This is an excellent manuscript dealing with very novel application of triboelectricity. Many triboelectric devices have been demonstrated ever since Prof. Wang introduced the tribo-electric energy harvesting idea many years ago. Since then this area has seen an unprecedented growth. In this work the authors demonstrate another novel and creative concept of nanoscale triboelectrification-gated transistor by using contact mode AFM and KPFM. The authors have systematically investigated the influence of tip voltage, contact force, scan number, and contact region on the transistor performance. The data presented in this manuscript are of excellent quality. The results are very intriguing, and I am sure that it will be of great interest to readers of Nature Comm. This concept has potential for furthering advances next generation electronics.

The manuscript is well prepared and well written. However, I have some minor comments and questions. The manuscript can be accepted after addressing the following comments:

- 1) It seems like the KPFM contrast saturates very fast as a function of contact force and scan number. Can the authors comment on this. Adding couple of sentences for clarification will help general readership.
- 2) Have the authors tried experiments with force lower than 1 nN to see the response trend?
- 3) It is also great idea to investigate scan speed. Have the authors conducted scan speed-dependent trend? If so, please add some comments for clarity.
- 4) From the application perspective, is there any optimal surface oxide value for the nanoscale triboelectrification-gated transistor?
- 5) Recently there have been reports about electron tunneling in tribo contacts when the thickness is extremely small (Liu, et al. Nature nanotechnology 13, 112 (2018); Nano Energy 48, 320 (2018); Matter 1, 650 (2019)). Is it possible to exploit tunneling concept in this arrangement. This may allow operation with extremely small currents thus increasing the energy efficiency in electronics (lower heat).

Reviewer #2 (Remarks to the Author):

The authors demonstrated triboelectrification-controlled electronics and established direct modulation mechanism by external mechanical stimuli. They studied a nanoscale triboelectrification-gated transistor (NTT) with contact-mode atomic force microscopy (AFM) and scanning Kelvin probe microscopy. However, the concept of triboelectrification-gated transistor (NTT) has been introduced by the coupling of contact electrification and electrostatic induction (ref. Wenbo Peng et al., ACS Nano 2016. 10, 4, 4395-4402) As well as, the study of manipulated nanoscale triboelectrification by AFM including the effects of contact cycles, contact region and charge diffusion on the characteristics of the dielectric surface has been well known (ref. Yu Sheng Zhou et al., Nano Lett. 2013. 13, 6, 2771-2776). This paper was well organized but has no novelty. Therefore, I am afraid I am not persuaded that these findings represent a sufficiently striking conceptual advance to justify publication in Nature Communications. For these reasons, I feel that these findings would be better suited for publication in an alternative journal.

--

It is necessary to prove that the gating effect by triboelectrification is sufficient to operate the p-Si channel. The author should check the following.

- All Id-Vd curves were not saturated. The author should check the data in the range of Vd over 5V.
- The bottom gate structure of Figure 2, p-Si / oxide gate insulator / p-Si structure, is not suitable for comparison with the concept of triboelectrification-gated transistor (NTT). The device performance according to gate bias should be compared with the top metal gate electrode based

on the same structure of Figure 5a. It is necessary to prove whether the role of the actual metal gate electrode can be replaced with triboelectrification-gating by checking the performance, mobility, Ion/off ratio, and V_{th} etc.

- In the inset of Figure 5e, it is necessary to show that I_d is saturated by increasing the tip voltage, especially the negative direction.

Revisions and Responses to Reviewers

Manuscript ID: NCOMMS-19-30650-T

Manuscript Type: Article

Title: Nanoscale Triboelectrification-Gated Transistor

Authors: Tianzhao Bu, Liang Xu, Zhiwei Yang, Xiang Yang, Guoxu Liu, Yuanzhi Cao, Chi Zhang*, Zhong Lin Wang*

Corresponding author: Chi Zhang, czhang@binn.cas.cn; Zhong Lin Wang, zlwang@gatech.edu

We are quite appreciative to the reviewers for the comments and valuable suggestions for our manuscript. Appended are the revisions that we made to the manuscript and supplementary information as well as the responses to each of the reviewer's comments. In the marked up revised manuscript and supplementary information, all the revisions are highlighted.

List of revisions made to the manuscript and supplementary information:

Rev 1. Page 2, line 28 and Page 3, line 2: electrostatic induction²⁴⁻²⁸; external mechanical stimuli²⁹⁻³²

Rev 2. Page 3, line 3-6: However, the interactive interfaces between external environment and electronics in current tribotronic devices are all in the macro scale, which has limited the integration and modularization of tribotronics. When the size scales down to the micro or nano range, whether the modulation effect still exists is a critical question for tribotronics.

Rev 3. Page 3, line 16-25: This work has experimentally realized the nanoscale triboelectric modulation on transistor by using AFM and demonstrated micro/nano-scale tribotronics for the first time, which could provide a deep understanding for the theoretical mechanism of tribotronics. The implementation of the NTT can provide direct interactions of electronics with external stimuli, which is highly desired for the development of micro/nano-electronics in diversity and

functionality. This may have great prospects in nanoscale transistor, micro/nano-electronic circuit and nano-electromechanical system (NEMS) for human-machine interfacing, flexible electronics, biomedical diagnosis/therapy and so on.

Rev 4. Page 5, line 21-29: The effect of contact force F on the characteristics of the NTT is first investigated. In this experiment, the top SiO₂ surface above the channel region was scanned by the AFM tip once at a scan speed of 10 μm/s. As shown in Figures S4a-c, the changed top SiO₂ surface potential above the channel region ΔV is enhanced with the contact force increasing from 0 to 2.5 nN, and approximately reaches a saturated value of -1.48 V when the contact force is larger than 2 nN. The corresponding drain current I_d of the NTT shows a similar trend following the increase of contact force, which is enhanced from 139 μA to 272 μA and reaches saturation when contact force is larger than 2 nN.

Rev 5. Page 5, line 29 and Page 6, line 1-9: During the process of nanoscale triboelectrification, the increasing contact force can decrease the potential barrier and induce more electrons to transfer onto the SiO₂ surface, until reaching the saturation state when the highest filled surface energy state of SiO₂ is as high as the Fermi level of Si. In the revised Figure S4, as the contact force increases in the low range, more charges are transferred that leads to the rise of the highest filled surface energy state of SiO₂. When the contact force increases to 2 nN, the amount of transferred charges is large enough to make the highest filled surface energy state of SiO₂ almost as high as the Fermi level of Si. So, the contact force larger than 2 nN will hardly lead to more transferred charges, and the KPFM contrast shows the fast saturation.

Rev 6. Page 6, line 10-20: Then we have investigated the effect of scan speed v on the characteristics of the NTT. In this experiment, the top SiO₂ surface above the channel region was scanned by the AFM tip once at a contact force of 1 nN. The changed top SiO₂ surface potential above the channel region ΔV is enhanced as the scan speed decreases from 16 μm/s to 2 μm/s, and reaches saturation when scan speed decreases to 4 μm/s (Figure S5a-c). The corresponding I_d output characteristics also shows an increasing trend with the decreasing of scan speed, and reaches a saturated value of

272 μA when the scan speed is less than 4 $\mu\text{m/s}$ (Figure S5d). A possible reason is that the decreasing scan speed could allow better contact between the AFM tip and the SiO_2 surface until a very low speed, enabling more charges to be transferred onto the SiO_2 surface.

Rev 7. Page 8, line 7-13: The increasing contact cycles can induce more transferred charges and leads to the rise of the highest filled surface energy state of SiO_2 .

According to Figure 2, when the SiO_2 surface is scanned twice by the AFM Si tip, the amount of transferred charges is large enough to make the highest filled surface energy state of SiO_2 almost as high as the Fermi level of Si. So, the increasing contact cycles (more than 2) will hardly lead to more transferred charges, and the KPFM contrast shows the fast saturation.

Rev 8. Page 11, line 13: -10 V to 10 V.

Rev 9. Page 11, line 14-16: The transfer curve of the NTT is saturated with the I_d of 125 μA in the positive direction, and in the negative direction, the drain current shows a saturation trend and is almost saturated with a value of 443 μA .

Rev 10. Page 11, line 19-27 and Page 12, line 1-8: In addition, we have fabricated a top metal gate electrode on the transistor (Figure S9a). The corresponding I_d - V_{tg} transfer curve is shown in Figure S9b, from which we can calculate the mobility ($\mu = \frac{Lg_m}{WC_{\text{ox}}V_d}$), where μ , g_m , L , W and C_{ox} are the mobility, transconductance, channel length, channel width and gate oxide capacitance), $I_{\text{on/off}}$ ratio, and V_{th} as 193 $\text{cm}^2\text{V}^{-1}\text{s}^{-1}$, 23 and 2.8 V, respectively. As for the NTT, the changed top SiO_2 surface potential above channel region ΔV is considered as the actual voltage that applied on the top gate of the NTT by the AFM tip. Thus, we get the I_d - ΔV transfer curve of the NTT to compare with the top metal gate transistor. According to the I_d - V_T curve shown in the inset of Figure 5e, the I_d - ΔV transfer curve of the NTT can be plotted as shown in Figure S9c, in which each ΔV corresponds to a tip voltage V_T (Figure 5d). The mobility, $I_{\text{on/off}}$ ratio, and V_{th} of the NTT can be further calculated as 431 $\text{cm}^2\text{V}^{-1}\text{s}^{-1}$, 4 and 1.1 V, respectively. As a contrast, there are a few differences on the mobility, $I_{\text{on/off}}$ ratio, and V_{th} between top metal gate transistor and the NTT due to the stray capacitance of top metal gate, the

residual charge after nanoscale triboelectrification and so on. However, both sets of values are comparable, which indicate that the top metal gate electrode can be replaced by nanoscale triboelectrification-gating.

Rev 11. Page 16, line 19-25 and Page 17, line 3-4:

26. Liu, J. *et al.* Direct-current triboelectricity generation by a sliding Schottky nanocontact on MoS₂ multilayers. *Nat. Nanotechnol.* **13**, 112–116 (2018).
27. Liu, J. *et al.* Sustained electron tunneling at unbiased metal-insulator-semiconductor triboelectric contacts. *Nano Energy* **48**, 320–326 (2018).
28. Liu, J. *et al.* Separation and Quantum Tunneling of Photo-generated Carriers Using a Tribo-Induced Field. *Matter* **1**, 650–660 (2019).
32. Xi, F. *et al.* Tribotronic bipolar junction transistor for mechanical frequency monitoring and use as touch switch. *Microsystems Nanoeng.* **4**, 25 (2018).

Rev 12. Page 19, line 1: Figure 2d has been revised.

Rev 13. Page 20, line 1: Figure 3d and Figure 3f have been revised.

Rev 14. Page 21, line 1: Figure 4d has been revised.

Rev 15. Page 22, line 1: Figure 5b, Figure 5c, Figure 5d, Figure 5e and Figure 5f have been revised.

Rev 16. Page 21, line 8-9: -10 V to 10 V.

Rev 17. Supplementary Information, Page 3, line 1: Figure S4 has been revised.

Rev 18. Supplementary Information, Page 3, line 4-8: b) Corresponding potential distributions in cross-sectional view with different contact forces. c) The potential difference between the rubbed and surrounding area with different contact forces. d) I_d output characteristics at a drain voltage of 5 V with different contact forces from 0 to 2.5 nN, and inset depicts the I_d - F transfer characteristics.

Rev 19. Supplementary Information, Page 4, line 1: Figure S5 has been added.

Rev 20. Supplementary Information, Page 4, line 2-8: **Figure S5. The effect of scan speed on the characteristics of the NTT.** a) Surface potential distributions of the NTT after regionally rubbed by the AFM tip with increasing scan speed. b) Corresponding potential distributions in cross-sectional view with different scan

speeds. c) The potential difference between the rubbed and surrounding area with different scan speeds. d) I_d output characteristics at a drain voltage of 5 V with different scan speeds from 16 to 2 $\mu\text{m/s}$, and inset depicts the I_d - v transfer characteristics.

Rev 21. Supplementary Information, Page 7, line 1: Figure S9 has been added.

Rev 22. Supplementary Information, Page 7, line 2-6: **Figure S9. Comparison between the top metal gate transistor and the NTT.** a) Top-view microscope images of the transistor i) without top metal gate electrode and ii) with top metal gate electrode. b) The corresponding I_d - V_{tg} transfer characteristics of the top metal gate transistor at a drain voltage of 5 V, and inset depicts the measurement circuit. c) The I_d - ΔV transfer curve of the NTT at a drain voltage of 5 V.

Responses to the reviewers:

Reviewer #1

This is an excellent manuscript dealing with very novel application of triboelectricity. Many triboelectric devices have been demonstrated ever since Prof. Wang introduced the tribo-electric energy harvesting idea many years ago. Since then this area has seen an unprecedented growth. In this work the authors demonstrate another novel and creative concept of nanoscale triboelectrification-gated transistor by using contact mode AFM and KPFM. The authors have systematically investigated the influence of tip voltage, contact force, scan number, and contact region on the transistor performance. The data presented in this manuscript are of excellent quality. The results are very intriguing, and I am sure that it will be of great interest to readers of Nature Comm. This concept has potential for furthering advances next generation electronics.

The manuscript is well prepared and well written. However, I have some minor comments and questions. The manuscript can be accepted after addressing the following comments:

Response: Thank you very much for your positive comments on the manuscript. We will make persistent efforts.

Q1: It seems like the KPFM contrast saturates very fast as a function of contact force and scan number. Can the authors comment on this? Adding couple of sentences for clarification will help general readership.

Response: Thank you for the suggestion. When the AFM Si tip is in contact with the surface of SiO₂ for nanoscale triboelectrification, electrons can flow from the AFM Si tip onto the SiO₂ surface to fill up the surface states, because the highest filled surface energy state of SiO₂ is below the Fermi level of Si^[1]. In this process, the increasing contact force can decrease the potential barrier and induce more electrons to transfer onto the SiO₂ surface, until reaching the saturation state when the highest filled surface energy state of SiO₂ is as high as the Fermi level of Si^[2]. In the revised Figure S4, as the contact force increases in the low range, more charges are transferred that leads to the rise of the highest filled surface energy state of SiO₂. When the contact force increases to 2 nN, the amount of transferred charges is large enough to make the highest filled surface energy state of SiO₂ almost as high as the Fermi level of Si. So, the contact force larger than 2 nN will hardly lead to more transferred charges, and the KPFM contrast shows the fast saturation.

Similarly, the increasing scan number can also induce more transferred charges and leads to the rise of the highest filled surface energy state of SiO₂. According to Figure 2, when the SiO₂ surface is scanned twice by the AFM Si tip, the amount of transferred charges is large enough to make the highest filled surface energy state of SiO₂ almost as high as the Fermi level of Si. So, the increasing scan number (more than 2) will hardly lead to more transferred charges, and the KPFM contrast shows the fast saturation. The relevant revisions are made in **Rev 5, 7, 17 and 18**.

Q2: Have the authors tried experiments with force lower than 1 nN to see the response trend?

Response: Thank you for your question. We have detailedly investigated the effect of contact force F on the characteristics of the NTT, including the contact forces lower than 1 nN. In this experiment, the top SiO₂ surface above the channel region was

scanned by the AFM tip once at a scan speed of 10 $\mu\text{m/s}$. As shown in Figures S4a-c, the changed top SiO_2 surface potential above the channel region ΔV is enhanced with the contact force increasing from 0 to 2.5 nN, and approximately reaches a saturated value of -1.48 V when the contact force is larger than 2 nN. The corresponding drain current I_d of the NTT shows a similar trend following the increase of contact force, which is enhanced from 139 μA to 272 μA and reaches saturation when contact force is larger than 2 nN. The relevant revisions are made in **Rev 4, 17 and 18**.

Figure S4. The effect of contact force on the characteristics of the NTT. a) Surface potential distributions of the NTT after regionally rubbed by the AFM tip with increasing contact force. b) Corresponding potential distributions in cross-sectional view with different contact forces. c) The potential difference between the rubbed and surrounding area with different contact forces. d) I_d output characteristics at a drain voltage of 5 V with different contact forces from 0 to 2.5 nN, and inset depicts the I_d - F transfer characteristics.

Q3: It is also great idea to investigate scan speed. Have the authors conducted scan

speed-dependent trend? If so, please add some comments for clarity.

Response: Thank you very much for the suggestion. We have investigated the effect of scan speed v on the characteristics of the NTT. In this experiment, the top SiO₂ surface above the channel region was scanned by the AFM tip once at a contact force of 1 nN. The changed top SiO₂ surface potential above the channel region ΔV is enhanced as the scan speed decreases from 16 $\mu\text{m/s}$ to 2 $\mu\text{m/s}$, and reaches saturation when scan speed decreases to 4 $\mu\text{m/s}$ (Figure S5a-c). The corresponding I_d output characteristics also shows an increasing trend with the decreasing of scan speed, and reaches a saturated value of 272 μA when the scan speed is less than 4 $\mu\text{m/s}$ (Figure S5d). A possible reason is that the decreasing scan speed could allow better contact between the AFM tip and the SiO₂ surface until a very low speed, enabling more charges to be transferred onto the SiO₂ surface. The relevant revisions are made in **Rev 6, 18 and 20.**

Figure S5. The effect of scan speed on the characteristics of the NTT. a) Surface potential distributions of the NTT after regionally rubbed by the AFM tip with increasing scan speed. b) Corresponding potential distributions in cross-sectional view

with different scan speeds. c) The potential difference between the rubbed and surrounding area with different scan speeds. d) I_d output characteristics at a drain voltage of 5 V with different scan speeds from 16 to 2 $\mu\text{m/s}$, and inset depicts the I_d - v transfer characteristics.

Q4: From the application perspective, is there any optimal surface oxide value for the nanoscale triboelectrification-gated transistor?

Response: Thank you for your question. The thickness of the top SiO_2 layer (d) is the main surface oxide parameter of the NTT. According to the classical theory of field effect transistor, the layer of oxide can be treated based on a parallel-plate capacitor model to analyze the effect of d , as shown in Figure R1. Generally, the transferred negative charges on the upper surface of the top SiO_2 layer will induce the same amount of positive charges on the interface between the P-Si and the SiO_2 layers. The generated electric field will concentrate in the dielectric layer, which is independent of the thickness d for an ideal parallel-plate capacitor model as below:

$$E_{\text{SiO}_2} = \frac{U_{\text{SiO}_2}}{d} = \frac{Q}{C_{\text{SiO}_2} d} = \frac{Q}{\epsilon_0 \epsilon_{\text{SiO}_2} S}$$

where U_{SiO_2} , C_{SiO_2} , ϵ_{SiO_2} and S are the voltage, capacitance, relative dielectric constant and area of the top SiO_2 layer, ϵ_0 is the vacuum dielectric constant, Q is the amount of transferred charges.

Actually, with increasing d , the capacitor will suffer from edge effect which causes leakage of electric field. Thus the induced positive charges at the interface will decrease. So for the practical NTT, decreasing d can suppress the leakage of electric field and enhance hole concentration at the interface, which is beneficial for the triboelectrification modulation effect. Nevertheless, the thickness of the top SiO_2 layer could not decrease to a too small value considering the tunneling effect. This could provide a general design guidance of the surface oxide value for the device fabrication. We will also detailedly analyze the effect of surface oxide value in our future work.

Figure R1. The schematic gate capacitance structure of the NTT with transferred charges on the surface.

Q5: Recently there have been reports about electron tunneling in tribo contacts when the thickness is extremely small (Liu, et al. Nature nanotechnology 13, 112 (2018); Nano Energy 48, 320 (2018); Matter 1, 650 (2019). Is it possible to exploit tunneling concept in this arrangement? This may allow operation with extremely small currents thus increasing the energy efficiency in electronics (lower heat).

Response: Thank you for your suggestion. It is an inspiring idea that using the tunneling concept in our arrangement. The electron tunneling is considered to be a small current which is appropriate to operate the current-control electronic device, for example bipolar junction transistor (BJT). According to a previous work (Xi, et al. Microsystems and Nanoengineering 2018, 4, 1, 25.), the electron tunneling in triboelectrification could serve as a base current to control the BJT. This may provide a solution for using the tunneling concept in tribotronics, which we will consider in our future works. In addition, we have cited the corresponding papers. The relevant revisions are made in **Rev 1 and 11**.

Reviewer #2

The authors demonstrated triboelectrification-controlled electronics and established direct modulation mechanism by external mechanical stimuli. They studied a nanoscale triboelectrification-gated transistor (NTT) with contact-mode atomic force microscopy (AFM) and scanning Kelvin probe microscopy. However, the concept of triboelectrification-gated transistor (NTT) has been introduced by the coupling of

contact electrification and electrostatic induction (ref. Wenbo Peng et al., ACS Nano 2016. 10, 4, 4395-4402) As well as, the study of manipulated nanoscale triboelectrification by AFM including the effects of contact cycles, contact region and charge diffusion on the characteristics of the dielectric surface has been well known (ref. Yu Sheng Zhou et al., Nano Lett. 2013.13, 6, 2771-2776). This paper was well organized but has no novelty. Therefore, I am afraid I am not persuaded that these findings represent a sufficiently striking conceptual advance to justify publication in Nature Communications. For these reasons, I feel that these findings would be better suited for publication in an alternative journal.

Response: Thank you very much for your comments. In the past years, many tribotronic devices have been demonstrated ever since the tribotronics was first proposed by the coupling of triboelectricity and semiconductor (ref. Chi Zhang et al., ACS Nano 2014. 8, 8, 8702-8709). However, the interactive interfaces between external environment and electronics in current tribotronic devices are all in the macro scale, which has limited the integration and modularization of tribotronics. When the size scales down to the micro or nano range, whether the modulation effect still exists is a critical question for tribotronics. As you have mentioned, the availability of triboelectrification-gated transistor has been predicted theoretically in 2016 (ref. Wenbo Peng et al., ACS Nano 2016. 10, 4, 4395-4402) by simulation. However, nothing has been done before our work to show that this is an experimentally achievable effect in micro/nano scale and the NTT can be realized substantially. Meanwhile, you have pointed that the nanoscale triboelectrification by AFM has been studied (ref. Yu Sheng Zhou et al., Nano Lett. 2013.13, 6, 2771-2776). This work has only studied the nanoscale triboelectrification characteristics by AFM, but not the modulation effect on the tribotronics which is much different.

In our work, we have experimentally realized the nanoscale triboelectric modulation on transistors by using AFM and demonstrated micro/nano-scale tribotronics for the first time. This could provide a deep understanding for the theoretical mechanism of tribotronics, by studying the effects of contact force, scan speed, contact cycle, contact region, charge diffusion and tip voltage. Moreover, the

implementation of the NTT can provide direct interactions of electronics with external stimuli, which is highly desired for the development of micro/nano-electronics in diversity and functionality. This may have great prospects in nanoscale transistor, micro/nano-electronic circuit and nano-electromechanical system (NEMS) for human-machine interfacing, flexible electronics, biomedical diagnosis/therapy and so on.

We hope these additional clarifications could address your concerns on the novelty of this work and receive your support. The manuscript has been substantially revised according to your suggestions. Especially, we have revised the introduction section to emphasize the novelty of this work. The relevant revisions are made in **Rev 2 and 3**.

It is necessary to prove that the gating effect by triboelectrification is sufficient to operate the p-Si channel. The author should check the following.

Q1: All I_d - V_d curves were not saturated. The author should check the data in the range of V_d over 5 V.

Response: Thank you for the suggestion. We have supplemented all the I_d - V_d output curves with drain voltage V_d increasing from 0 to 20 V. Each I_d - V_d output curve has a saturation area as shown in Figure R2. The relevant revisions are made in **Rev 11, 13, 14 and 15**.

Figure R2. a) I_d - V_d output characteristics with different contact cycles. b) I_d - V_d output characteristics with different contact lengths. c) I_d - V_d output characteristics with different contact widths. d) I_d - V_d output characteristics with different dissipation times. e) I_d - V_d output characteristics with different tip voltages.

Q2: The bottom gate structure of Figure S2, p-Si / oxide gate insulator / p-Si structure, is not suitable for comparison with the concept of triboelectrification-gated transistor (NTT). The device performance according to gate bias should be compared with the top metal gate electrode based on the same structure of Figure 5a. It is necessary to

prove whether the role of the actual metal gate electrode can be replaced with triboelectrification-gating by checking the performance, mobility, $I_{on/off}$ ratio, and V_{th} etc.

Response: Thank you very much for the suggestion. We have fabricated a top metal gate electrode on the transistor (Figure S9a). The corresponding I_d - V_{tg} transfer curve

is shown in Figure S9b, from which we can calculate the mobility ($\mu = \frac{Lg_m}{WC_{ox}V_d}$),

where μ , g_m , L , W and C_{ox} are the mobility, transconductance, channel length, channel width and gate oxide capacitance), $I_{on/off}$ ratio, and V_{th} as $193 \text{ cm}^2\text{V}^{-1}\text{s}^{-1}$, 23 and 2.8 V, respectively. As for the NTT, the changed top SiO_2 surface potential above channel region ΔV is considered as the actual voltage that applied on the top gate of the NTT by the AFM tip. Thus, we get the I_d - ΔV transfer curve of the NTT to compare with the top metal gate transistor. According to the I_d - V_T curve shown in the inset of Figure 5e, the I_d - ΔV transfer curve of the NTT can be plotted as shown in Figure S9c, in which each ΔV corresponds to a tip voltage V_T (Figure 5d). The mobility, $I_{on/off}$ ratio, and V_{th} of the NTT can be further calculated as $431 \text{ cm}^2\text{V}^{-1}\text{s}^{-1}$, 4 and 1.1 V, respectively. As a contrast, there are a few differences on the mobility, $I_{on/off}$ ratio, and V_{th} between top metal gate transistor and the NTT due to the stray capacitance of top metal gate, the residual charge after nanoscale triboelectrification and so on. However, both sets of values are comparable, which indicate that the top metal gate electrode can be replaced by nanoscale triboelectrification-gating. The relevant revisions are made in **Rev 10, 21 and 22.**

Figure S9. Comparison between the top metal gate transistor and the NTT. a) Top-view microscope images of the transistor i) without top metal gate electrode and ii) with top metal gate electrode. b) The corresponding I_d - V_{tg} transfer characteristics of the top metal gate transistor at a drain voltage of 5 V, and inset depicts the measurement circuit. c) The I_d - ΔV transfer curve of the NTT at a drain voltage of 5 V.

Q3: In the inset of Figure 5e, it is necessary to show that I_d is saturated by increasing the tip voltage, especially the negative direction.

Response: Thank you for the suggestion. We have revised the I_d - V_T transfer curve with V_T from -10 V to 10 V. The revised curve is saturated with the I_d of 125 μA in the positive direction. Meanwhile, in the negative direction, the drain current shows a saturation trend and is almost saturated with a value of 443 μA. The relevant revisions are made in **Rev 7, 8, 14 and 15**.

The inset of **Figure 5e** depicts the I_d - V_T transfer characteristics with tip voltages from -10 V to 10 V.

References

1. Lowell, J. & Rose-Innes, A. C. Contact electrification. *Adv. Phys.* **29**, 947–1023 (1980).
2. Zhou, Y. S. *et al.* Manipulating nanoscale contact electrification by an applied electric field. *Nano Lett.* **14**, 1567–1572 (2014).

REVIEWERS' COMMENTS:

Reviewer #1 (Remarks to the Author):

The authors have addressed all the questions satisfactorily. Publish as it is in the current revised version.

Reviewer #2 (Remarks to the Author):

Dear Editor

The authors clearly responded to the reviewer's comments. And, the modified manuscript is well organized. I recommend publication in Nature Communications for the paper in its present form.

Responses to Reviewers

Manuscript ID: NCOMMS-19-30650A

Manuscript Type: Article

Title: Nanoscale Triboelectrification-Gated Transistor

Authors: Tianzhao Bu, Liang Xu, Zhiwei Yang, Xiang Yang, Guoxu Liu, Yuanzhi Cao, Chi Zhang*, Zhong Lin Wang*

Corresponding author: Chi Zhang, czhang@binn.cas.cn; Zhong Lin Wang, zlwang@gatech.edu

Responses to the reviewers:

Reviewer #1

The authors have addressed all the questions satisfactorily. Publish as it is in the current revised version.

Response: Thank you very much for your positive comments. We will make persistent efforts.

Reviewer #2

The authors clearly responded to the reviewer's comments. And, the modified manuscript is well organized. I recommend publication in Nature Communications for the paper in its present form.

Response: Thank you very much for your positive comments. We will make persistent efforts.